# The cGMP Pathway and Inherited Photoreceptor Degeneration: Targets, Compounds, and Biomarkers

**DOI:** 10.3390/genes10060453

**Published:** 2019-06-14

**Authors:** Arianna Tolone, Soumaya Belhadj, Andreas Rentsch, Frank Schwede, François Paquet-Durand

**Affiliations:** 1Institute for Ophthalmic Research, University of Tübingen, Elfriede-Aulhorn-Strasse 5-7, 72076 Tübingen, Germany; arianna.tolone@uni-tuebingen.de (A.T.); soumaya.belhadj@uni-tuebingen.de (S.B.); 2Biolog Life Science Institute, 28199 Bremen, Germany; ar@biolog.de (A.R.); fs@biolog.de (F.S.)

**Keywords:** retina, cyclic GMP, apoptosis, necrosis, drug delivery systems, translational medicine

## Abstract

Photoreceptor physiology and pathophysiology is intricately linked to guanosine-3’,5’-cyclic monophosphate (cGMP)-signaling. Here, we discuss the importance of cGMP-signaling for the pathogenesis of hereditary retinal degeneration. Excessive accumulation of cGMP in photoreceptors is a common denominator in cell death caused by a variety of different gene mutations. The cGMP-dependent cell death pathway may be targeted for the treatment of inherited photoreceptor degeneration, using specifically designed and formulated inhibitory cGMP analogues. Moreover, cGMP-signaling and its down-stream targets may be exploited for the development of novel biomarkers that could facilitate monitoring of disease progression and reveal the response to treatment in future clinical trials. We then briefly present the importance of appropriate formulations for delivery to the retina, both for drug and biomarker applications. Finally, the review touches on important aspects of future clinical translation, highlighting the need for interdisciplinary cooperation of researchers from a diverse range of fields.

## 1. The Retina and Inherited Photoreceptor Degeneration

The retina transforms photons of light in electro-chemical signals, processes these signals, and transmits light-dependent information to different areas of the central nervous system [1]. The human retina is affected by a large number of hereditary, typically monogenic diseases, causing severe vision impairment or blindness [2,3]. Genetic diseases causing the degeneration and loss of the light-sensitive photoreceptors in the retina are grouped under the term inherited retinal degeneration (IRD) [4,5]. Photoreceptor loss in IRD-type diseases in most cases remains untreatable, making it a major unresolved medical problem [2,6]. This review focuses on the guanosine-3’,5’-cyclic monophosphate (cGMP)-signaling pathway and its role in IRD and how the components of this pathway may be exploited both for the development of new therapies as well as for new biomarkers for the evaluation of treatment efficacy. 

In IRD-type diseases, the causative genetic defect typically causes a primary degeneration of rod photoreceptors (rods), with subsequent, secondary loss of cone photoreceptors (cones), eventually leading to complete blindness [4]. The result is an almost complete loss of the outer nuclear layer (ONL) and outer plexiform layer of the retina, while the inner retina remains intact initially (Figure 1). However, eventually in the inner retina the dendrites of bipolar and horizontal cells retract and an extensive gliotic scar may form [7]. The secondary degeneration of cones can be a surprisingly slow process, with some cones surviving for many years beyond the main degeneration phase [8]. 

The IRD group of diseases is characterized by a vast genetic heterogeneity, with disease-causing mutations known in over 270 genes (https://sph.uth.edu/retnet; information retrieved April 2019). This diversity severely hinders both the understanding of degenerative mechanisms and the development of treatments. To complicate matters further, each of these IRD-linked genes may carry many different types of either recessive or dominant mutations, ranging from complete loss-of-function to gain-of-function [3]. At present, a rough estimate will put the total number of disease mutations to amount to at least several tens of thousands. This enormous heterogeneity calls for the development of treatment approaches targeting common mechanisms, that can effectively treat this condition regardless of the underlying genetic causes. 

## 2. cGMP-Signaling in Phototransduction and Degeneration

The physiology of photoreceptors and the phototransduction cascade critically depends on the signaling of the second messenger molecule cGMP [9]. Mutations affecting genes related to the phototransduction cascade often cause a dysregulation of cGMP, triggering a series of down-stream processes, which eventually kill photoreceptors [10,11]. This highlights cGMP-signaling as one principle common to many disease-causing mutations and, hence, as a plausible target for therapeutic interventions. 

In the phototransduction cascade, in the dark, high levels of cGMP in photoreceptor outer segments (OS) allow for the sensitization of photoreceptor cells down to the level of single-photon sensitivity [12]. cGMP binds to and opens the prototypic phototransduction target, the cyclic nucleotide-gated channel (CNGC), located in the outer membrane of the photoreceptor OS [13]. The CNGC opening allows for an influx of Na^+^ and Ca^2+^ into the OS, yet, at the same time K^+^ and Ca^2+^ ions are constantly extruded via the Na^+^/Ca^2+^/K^+^ exchanger (NCKX). This continuous influx and outflow of ions in the absence of light is referred to as the dark current [14]. The conformational change in the rhodopsin protein brought about by the absorption of a photon of light, leads to the sequential activation of the G-protein transducin and the phosphodiesterase-6 (PDE6). PDE6, which is located to the membranous disks within photoreceptor OS, hydrolyses cGMP, leading to the closure of CNGC and a hyperpolarization of the OS due to continued activity of NCKX. This in turn promotes the generation of an electro-chemical signal that is transmitted to second order neurons [15]. 

The toxicity of high levels of cGMP for photoreceptors was already established by Deborah Farber and Frank Lolley in the 1970s [10,16]. The discovery of disease-causing mutations in the PDE6 α and β genes [17,18] provided an explanation for excessive photoreceptor cGMP levels. Many more disease mutations are nowadays known to be associated with high cGMP-dependent photoreceptor cell death, including mutations in the genes encoding for CNGC [19], rhodopsin [11], AIPL1 [20], and photoreceptor guanylyl cyclase [21]. However, cGMP elevation was also found in cells with mutations in genes that seem to have no obvious connection to cGMP metabolism. An example for this situation are mutations in the *PRPH2* gene [22], which encodes for an OS structural protein [23]. 

Overall, high cGMP and cGMP-dependent cell death are likely involved in a significant proportion of IRD patients [24], making it an attractive target for therapeutic interventions, and additionally highlighting it, or its downstream processes, for biomarker development. 

## 3. Targeting cGMP-Signaling 

cGMP acts as a second messenger and plays a critical role in the regulation of different processes in many organisms. Cyclic nucleotide research began in the 1960s but the biological role of cGMP was identified only in the 1980s thanks to two important discoveries: the cGMP synthesis stimulation by the atrial natriuretic peptide (ANP) in the heart, and the cGMP synthesis stimulation by nitric oxide (NO) in smooth muscle cells causing vasorelaxation [25]. In the retina, cGMP was identified in the 1970s when the presence of high activities of guanylate cyclase, as well as a protein kinase stimulated by cGMP, were described in the OS of bovine rods [26]. Today, we know that cGMP, when localized to the photoreceptor OS, is an essential component of the phototransduction cascade [9]. However, cGMP also has targets outside the phototransduction cascade, notably protein kinase G (PKG; also referred to as cGMP-dependent protein kinase, cGK), which appears to be highly relevant for photoreceptor cell death [22].

### 3.1. Regulation of Photoreceptor cGMP Synthesis

The synthesis of cGMP is catalyzed by membrane guanylyl cyclases (GCs), which convert guanosine 5´-triphosphate (GTP) into cGMP. Photoreceptor GCs work differently compared to other membrane GCs: They do not respond to extracellular ligands, but instead are regulated by Ca^2+^-binding, and GC activating proteins (GCAPs) [27]. GCAPs are proteins containing EF-hand motifs and once these motifs are occupied by Ca^2+^ they inhibit cGMP production. In the darkness, when intracellular Ca^2+^ is relatively high, photoreceptor guanylyl cyclases (RetGC1 and RetGC2) are inhibited by GCAPs and do not synthesize cGMP. Illumination induces rhodopsin conformational changes, which enable the activation of transducin, a GTP-binding protein. Activated transducin disinhibits PDE6, thereby allowing the hydrolysis of cGMP and the closure of CNGC leading to the interruption of Ca^2+^ influx. Since Ca^2+^ is constantly extruded via NCKX, CNGC closure quickly lowers intracellular Ca^2+^ levels. In this situation, Mg^2+^ replaces the Ca^2+^ bound to GCAPs, activates RetGCs, and promotes the synthesis of cGMP [27,28,29] (Figure 2). 

In this way, both cGMP and Ca^2+^ concentrations in the photoreceptor are closely linked by a feedback loop that would normally limit the levels of both second messengers to their physiological ranges [30]. The reasons why this feedback loop fails in IRD mutations are not currently known. A major focus of IRD research in the past was on the role of CNGC and the Ca^2+^ influx that it mediates, and a number of studies suggested that excessive activation of CNGC and too high intracellular Ca^2+^ levels would drive photoreceptor degeneration [31,32,33]. However, other, more recent studies failed to show a clear causal connection between Ca^2+^ and photoreceptor loss in IRD [15,34,35,36]. Therefore, in the following section, we focus on the role of PKG in photoreceptor degeneration and how this may be exploited for IRD therapy developments. 

### 3.2. The Role of PKG in Normal Physiology

Elevation of cGMP intracellular concentration induces a binding-dependent activation of PKG, a serine/threonine-specific protein kinase that phosphorylates a number of biological targets [37]. Mammals express two different genes for PKG: the *PRKG1* gene encodes for the Iα and Iβ isoforms of PKG I and is located on human chromosome 10, whereas the *PRKG2* gene encodes PKG II and is located on human chromosome 4. PKG is a homodimer of two identical subunits and each PKG subunit consists of an N-terminal regulatory domain, an autoinhibitory sequence, two tandem cGMP binding sites, and a C-terminal catalytic domain [38]. When cGMP binds to PKG it promotes a conformational change, which liberates the catalytic site. In mammals, PKG I mediates many of the effects of NO/cGMP on vasodilation, vascular smooth muscle cell relaxation, proliferation, and apoptosis. PKG II regulates homeostasis of Na^+^, Cl^−^, endochondral ossification of bones, and various functions of the nervous system [39]. Not much is known about the role of PKG in retinal photoreceptor cells and its significance for the regulation of phototransduction. Immunohistochemical staining for PKG I and PKG II isoforms revealed a predominant expression of PKG I in photoreceptors ONL [40], while PKG II seems to be expressed in the inner nuclear layer (INL) and the ganglion cell layer (GCL) [41]. 

Important insight into the physiological functions of PKG isoforms also came from genetic knock-out animals. A full knockout of PKG I (α and β) leads to defects in smooth muscles, intestinal dysfunctions, growth defects, and dwarfism [42]. Interestingly, this effect could be rescued by the reconstitution (knock-in) of the murine PKG (cGK) Iα or Iβ isozymes in smooth muscles [43] allowing normal development. In sensory cells of the inner ear, a knock-out of the murine *Prkg1* gene did not affect hearing function in detectable ways, yet, *Prkg1* knock-out animals were protected against acoustic trauma and resistant against noise induced hearing loss [44]. When *Prkg2* was knocked-out in mice, the animals appeared generally healthy but showed a defect in the entrainment of circadian rhythm, even though retinal morphology and function appeared normal [45]. It remains unclear whether PKG II deficiency leads to changes in the photosensitive retinal ganglion cells. Overall, the results obtained on animals deficient for PKG enzymes [37] suggest that targeted inhibition of PKG would unlikely produce major adverse effects.

The idea that PKG activity plays a role in cell death has been widely established. For instance, activation of PKG has been used for the induction of apoptosis in colon cancer cells and in human breast cancer cells [46,47], and is linked to pro-apoptotic effects in ovarian cancer [48]. Excessive activation of PKG has been shown to cause cell death in certain neuronal cell types [49,50]. Several studies pointed to the importance of the cGMP/PKG-dependent cell death mechanism in photoreceptor degeneration and demonstrated the existence of a non-apoptotic cell death mechanisms involving cGMP-dependent over activation of PKG [11,22,51]. This evidence makes PKG a potential target for neuroprotective strategies. 

### 3.3. Effects of PKG Inhibition 

The different physiological functions of PKG in different tissues suggest that its inhibition can lead to various effects. For example, PKG inhibitors DT-2 and DT-3 have been demonstrated to decrease NO-mediated vasodilation in isolated cerebral arteries [52]. Inhibitors of PKG could; therefore, be used to antagonize vasoplegia, a hypotonic condition characteristic of anaphylactic shock [53]. Selective inhibition of PKG Iα mediated by the balanol-derivative N46 reduced thermal hyperalgesia and osteoarthritic pain in rats [54]. 

PKG inhibition in cancer treatment appears to have an ambiguous role. In some forms of cancer, the effects seem to be very positive. For example, in colorectal carcinoma (CRC) cell lines inhibition of NO/PKG/extracellular-signal-regulated kinases (ERK)-signaling mediated by the PKG-inhibitor KT5823 reduced migration and invasion in both scratch wound and modified Boyden chamber assays [55]. PKG seems to exert a pro-apoptotic role in breast cancer cell lines [46]. Some of the PKG inhibitors used in the experiments mentioned above may suffer from lack of specificity and potency. For instance, KT5823 is an ATP-binding site inhibitor and works efficiently as a PKG inhibitor in vitro [56]. However, it is also a weak inhibitor of PKC and PKA, both of which may constitute potential competitive binding sites. [56,57]. Another PKG inhibitor, DT-2, a substrate-binding site inhibitor, has been demonstrated to be inefficient in inhibiting PKG activity in several types of cells. Furthermore, in whole-cell homogenates the PKG specificity over PKA for DT-2 appears to be lost [58]. 

To understand PKG cellular functions as well as the cGMP/PKG-dependent cell death mechanisms, inhibitors or activators with very high selectivity are needed. A possible and valid alternative is the use of cGMP analogues. 

### 3.4. Design of cGMP Analogues to Inhibit PKG

Analogues of cGMP are a class of second messenger compounds able to inhibit or activate both PKG isoforms (Iα, Iβ, and II) [59] and have been used in a multitude of research areas. Over the last ~40 years, the first generation of cGMP analogues have become standard tools for investigations of biochemical and physiological signal transduction pathways [60]. More recently, newly developed PKG activators have been shown to reduce proliferation in colon cancer cell lines [61], as well as in melanoma cells [62]. The use of PKG inhibitors contributed to identify a common non-apoptotic cGMP-dependent degeneration mechanism in different animal models for IRD [22]. In particular, cGMP analogues used as PKG inhibitors carry the common motif of an Rp-configured phosphorothioate. Therein, sulfur replaces one of the exocyclic oxygens of the 3’,5’-cyclic phosphate (equatorial (eq) position in Figure 3), while the prefix “Rp” defines the configuration of this moiety according to Cahn–Ingold–Prelog rules for chiral atoms. These compounds, further referred to as Rp-cGMPS analogues, are able to bind to the cGMP-binding domains of PKGs, but do not evoke the conformational changes of the enzyme required for activation of the catalytically active C-subunit [63]. This leads to a competitive and reversible inhibition of the various PKG isoforms. Furthermore, Rp-cGMPS analogues were shown to be resistant against hydrolysis by mammalian 3’,5’-cyclic phosphodiesterases (PDEs) [64], the enzyme superfamily that metabolizes cGMP to 5’-GMP in vivo. Due to the sulfur in the cyclic phosphate and other substituents, such as bromine, mainly introduced to the guanine nucleobase, corresponding Rp-cGMPS analogues show improved lipophilicity leading to enhanced membrane permeability compared to cGMP [53,60]. Schematic structural motifs of Rp-cGMPS analogues with improved inhibitory potencies for PKGs are illustrated in Figure 3. Taken together, three factors are responsible for the efficacy of sophisticated Rp-cGMPS analogues in biological systems: 1) The improved stability against degradation, 2) the lipophilicity and membrane permeability, and 3) the inhibitory potential for PKG.

As mentioned above, cGMP-signaling is central to phototransduction, but cGMP levels elevated by excessive production or, conversely, by insufficient hydrolysis may trigger photoreceptor degeneration. Between 2012 and 2016, the DRUGSFORD consortium generated over 80 novel cGMP analogues that target and inhibit PKG, and showed that systemic intraperitoneal injections of a selected liposome-formulated Rp-cGMPS analogue can protect photoreceptors from degeneration in different animal models for IRD. This pharmacological treatment significantly preserved rod and—indirectly—cone photoreceptors, ensuring the maintenance of retinal function [65]. This proof of concept study also highlighted the versatility of cGMP(S) analogues, which can be designed and formulated both to study the cellular functions of PKG and as new lead compounds for therapeutic applications in IRD. 

## 4. Pre-Clinical and Clinical Biomarkers for Inherited Retinal Degeneration

Neurodegenerative diseases are often characterized by a slow progression of neuronal cell loss, sometimes over the course of several decades. This implies that disease symptoms may appear only when the damage is advanced and thus the diagnosis is made at a late stage of the disease. The current lack of effective treatment options for most neurodegenerative diseases, including those of the retina, stems in part from a lack of biomarkers to detect neuronal cell death and to study its progression and dynamics. In neurodegenerative diseases biomarkers are needed, not only to aid diagnosis and monitor disease progression, but also, as new medicines are introduced, to detect the patient’s response to treatment [68]. 

### 4.1. Review of Recent Retinal Biomarker Developments

Over the past decade there has been some progress in the identification and development of biomarkers for retinal cell death, based on the increasing knowledge of the underlying mechanisms behind retinal degeneration. Notably, several attempts have been made to develop molecular probes that target specific degenerative processes [41,69,70].

Perhaps the most straight-forward approach for biomarker development in terms of clinical applicability and patient friendliness would be biomarkers that could be analyzed in blood samples. While to date there are no established biochemical blood markers for retinal degeneration, a study published by Martinez-Fernandez de la Camara et al. [71] showed that in a group of patients affected by retinitis pigmentosa (RP), the serum cGMP levels are increased by approximately 65%, compared to a control group. These results were confirmed in an independent study on a family with autosomal recessive RP due to homozygous mutations in the *PDE6A* gene [72]. The latter study also found cGMP plasma concentration to have a good sensitivity, which means it can detect diseased subjects well, but with a lower specificity, thus carrying the risk that healthy subjects could appear as having the disease. Nevertheless, this example nicely illustrates the potential for the development of blood borne biomarkers. On the other hand, given the high genetic heterogeneity of IRD-type diseases, it is unlikely that a single blood-borne biomarker can be used for all patients. Most likely different mutations, in different genes, will require separate biomarker developments. In the end, this could lead to assays that consider maybe a few dozen different metabolic markers for IRD identification.

Photoreceptors are susceptible to oxidative stress because of their high metabolic rate as well as their exposition to environmental factors, such as ultraviolet radiation or high oxygen tension. Hence, it has been suggested that oxidative stress contributes to the pathogenesis of retinal degeneration [73]. In line with this concept, Martinez-Fernandez de la Camara and colleagues [71] determined the levels of different markers of the antioxidant-oxidant status in aqueous humor and blood from RP patients, and compared them with those in healthy controls to confirm an alteration of this status. The authors found reduced superoxide dismutase (SOD3) activity in aqueous humor as well as reduced SOD3 activity and increased TBARS (thiobarbituric acid reactive substances) formation in peripheral blood. Even though these studies are promising it remains to be seen whether the specificity and sensitivity of these parameters is sufficient for clinical biomarker applications. 

Another strategy to assess retinal cell death is to detect and quantify substances released by dying cells in ocular fluids. For instance, neurofilaments (Nf) are essential constituents of the axonal cytoskeleton [74] and, during neuronal cell death and axonal degeneration in the retina, contents of the cytoplasm such as Nf are released into the extracellular fluid [75]. From there, Nf diffuse into the vitreous body and anterior chamber fluid, which are the two compartments adjacent to the retina. Accordingly, in patients who underwent vitrectomy for retinal detachment, epiretinal gliosis, or macular hole surgery, the neurofilament heavy-chain protein could be quantified from the human vitreous body using an ELISA technique [75]. This could make Nf release a potential biomarker for retinal degeneration, provided that vitreous samples can be obtained.

A non-invasive detection of retinal cell death was suggested in a study published by Cordeiro and colleagues [76]. They established a proof-of-principle in reporting the use of fluorescent cell death markers to temporally resolve and quantify the early and late phases of apoptosis and necrosis of single nerve cells in glaucoma and Alzheimer’s disease animal models. More specifically, they used Alexa Fluor 488-labelled Annexin V to identify apoptotic cells, as well as propidium iodide to identify necrotic and late apoptotic cells. Fluorescence was visualized using confocal scanning laser ophthalmoscopy. This initial animal study eventually led to a phase 1 clinical trial [69], which established a proof-of-concept demonstrating that retinal cell death can be identified in the human retina with increased levels of activity in glaucomatous neurodegenerative disease.

### 4.2. Using the cGMP Pathway for Biomarker Development 

Cell death in IRD, and in other neurodegenerative diseases, is often thought to be governed by apoptosis, a form of cell death that would display a number of characteristic markers, such as cytochrome c leakage and activation of caspase-type proteases [77]. However, for photoreceptor degeneration in the retina, a growing body of evidence suggests the involvement of non-apoptotic, alternative cell death mechanisms [11,78]. In many cases photoreceptor cell death appears to be driven by cGMP-dependent activation of CNGC [19] and may be even more prominently by PKG [22], also in the absence of functional CNGC expression [79]. A number of cell death-related processes have been discovered down-stream of cGMP-signaling, including excessive activation of histone-deacetylase [80], calpain-type proteases [81], DNA-methyl-transferase [82] and poly-ADP-ribose-polymerase (PARP) [83,84]. Molecular probes targeting these enzymes could be developed and used as biomarkers to detect cell death and study its progression and dynamics in IRD. For instance, an assay previously adapted to resolve PARP activity in photoreceptors ex vivo [41], and which relied on the incorporation of biotin labelled NAD^+^ residues as subunits in elongated poly-ADP-ribose chains (Figure 4), could potentially be used to this effect. While the original assay required a two-step detection, using, first, biotinylated NAD^+^ and then fluorescently-labelled avidin, in the future it may be possible to develop a single step assay. This might then be used for direct in vivo detection, possibly via fluorescent scanning laser ophthalmoscopy [69].

Similar in vivo biomarkers for the cellular detection of retinal cell death might be developed in the future, using molecular probes targeting cGMP-dependent processes, such as calpain- or caspase-type proteases, histone deacetylases, or DNA-methyl-transferases. 

## 5. Delivery of Compounds to the Retinal Photoreceptors

Compounds targeting photoreceptor cGMP-signaling may have strong therapeutic potential and can, at least in part, address the genetic heterogeneity of IRD-type diseases. Nevertheless, to develop them into successful IRD treatments still requires the delivery of these compounds to the photoreceptor cell, which in most cases will entail formulating the active compound in a suitable drug delivery system (DDS).

### 5.1. Drug Delivery Systems (DDSs) for the Retina

The retina is protected from detrimental external agents (e.g., toxins, pathogens) by the blood–retinal barrier (BRB), as well as other ocular barriers that prevent therapeutic agents from reaching the photoreceptor cells in the retina [85]. To overcome this obstacle, a variety of different technical approaches have been pursued, using different routes of administration, such as suprachoroidal injection [86], subretinal injection [87], injection into the capsule of Tenon [88], and intravitreal injection [89]. Each of these administration routes may require the use of an adapted DDS such as glutathione conjugated liposomes [65]. Moreover, each of these routes have specific advantages and disadvantages, but which ever administration route is chosen the drug formulation/delivery system used will be critical to successful treatment development.

The use of different application (injection) routes in current clinical practice may be illustrated by two examples: In gene therapy, the most frequently used technique (mostly in clinical trial but also –since 2018–for FDA/EMA approved *RPE65* gene therapy) is subretinal injection of gene constructs delivered in adeno-associated-virus [90,91]. The procedure entails a detachment of the retina from the RPE with the risk for significant and irreversible retinal damage [87]. However, since gene therapy interventions are considered to be needed only once in a lifetime, the overall risk–benefit ratio for subretinal injection is deemed acceptable [90]. For anti-vascular endothelial growth factor (VEGF) medication, commonly used for the treatment of age-related-macular degeneration (AMD) and other retinal diseases, the most frequently used route of administration is intravitreal injection [92]. This form of application is easier to perform than subretinal injection and the risk for damage to the retina is far lower. On the other hand, anti-VEGF drugs require regular re-administration leading to a risk for cumulative damage to the eye, notably intraocular infections [89]. 

While the last ten years have seen an important development and the appearance of many innovative materials, designs, and technologies for retinal drug delivery, efficient and sustained drug delivery to the photoreceptors remains a major challenge. Importantly, each compound or therapeutic agent may require highly-adapted DDS, which additionally must comply with regulatory requirements from the medicinal drug and product authorities [93]. Therefore, future research into new treatments for IRD should take the retinal delivery problem into consideration as early as possible, and synchronize the compound and delivery system development. 

### 5.2. Retinal Drug Delivery: Local vs. Systemic Administration 

Currently, the standard route of administration for retinal drugs, such as anti-VEGF medications, is multiple dosing through intravitreal injection [89,94], while in the case of single-dose, gene therapeutic agents subretinal injection is the preferred technique [95]. Among the advantages of these local administration routes are the limited exposure of the whole body, not the least since the eye is an encapsulated organ. When a treatment does eventually leak out into the systemic circulation, even very high intraocular doses will be diluted about 10.000-fold (volume of the human eye ≈ 6.5 ml; volume of the human body (70 kg) ≈ 66 l; note that this dilution factor will be smaller in children). Therefore, the risk of systemic side-effects of intraocularly applied drugs appears rather small, even though such effects have been reported in some cases after anti-VEGF medication [92]. 

Among the disadvantages of intraocular application are the patients´ discomfort, the need for qualified doctors to perform the injections, and the low (but cumulative) risk for intraocular inflammation [96]. Unfortunately, topical drug application is usually not possible because of the various barriers surrounding the eye and retina, and the alternating lipophilic and hydrophilic nature of these barriers, preventing most drugs from reaching the retina via this route [93]. 

Systemic drug application might be a more patient-friendly, alternative administration route, yet, for the reasons laid out above, systemic application will require dramatically higher dosing to reach comparable intraocular drug concentration. In addition, cyclic nucleotide analogues are rapidly excreted via the renal system, strongly reducing bioavailability [65,97]. Both problems could potentially be circumvented if a drug intended for the retina was combined with a targeted delivery formulation, such as glutathione (GSH) conjugated liposomes [98]. Such liposomes were recently shown to allow for sufficient drug delivery of PKG inhibitors to the photoreceptors of the retina to obtain a significant morphological and functional rescue in animal models for IRD [65]. Nevertheless, given the potential for systemic adverse effects and the strongly increased consumption of the (often expensive) drug substance, it remains to be seen whether the systemic administration route will be viable in future clinical settings. 

## 6. Conclusions and Outlook

The cGMP-signaling system offers new opportunities for the development of innovative treatments for IRD, as well as for the development of biomarkers that can assess treatment efficacy. Novel small molecule analogues of cGMP allow PKG (and/ or CNGC) to be targeted, with high specificity, and have shown to achieve marked photoreceptor protection in different animal models [22,65]. While this was true for inhibitory cGMP analogues in forms of IRD connected to high photoreceptor cGMP levels, it remains to be seen whether activator analogues would have protective capacity in situations where photoreceptor cGMP is too low (e.g., in RetGC loss-of-function mutations [99]). 

Nevertheless, translation of new drugs or biomarkers based on cGMP-signaling, and their corresponding DDS, will require substantial development efforts, notably in the areas of ocular pharmacokinetics, GMP manufacturing, safety and tolerability testing, clinical trial design, and suitable clinical endpoints, etc. All these developments require expert knowledge and a broad interdisciplinary oversight, which, unfortunately is rarely found in both basic and clinical researchers. This highlights the need for specific educational programs to train scientists in translational research. Another critical aspect for successful translation into clinical practice is the progression into a viable commercial product that can recoup the development costs and sustain further development and improvements. Additionally, here a specific training and a sensitization of scientists for the requirements of commercialization (e.g., protection of intellectual property, appropriate documentation, regulatory requirements, business models in the rare disease space, etc.) will be important. 

## Figures and Tables

**Figure 1 genes-10-00453-f001:**
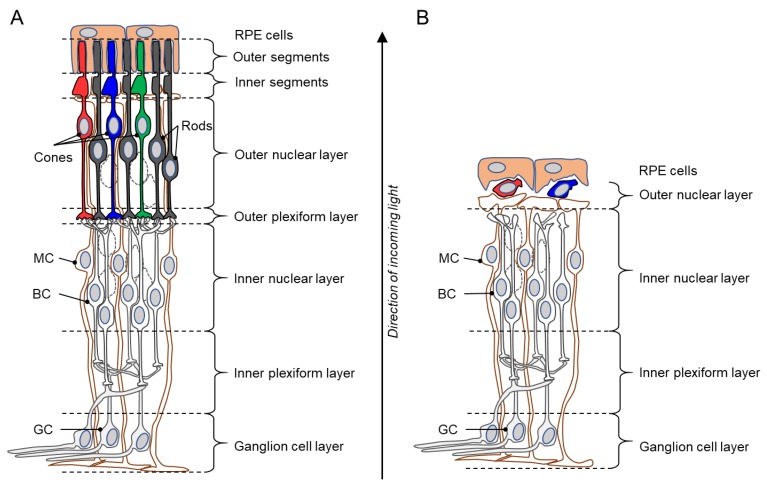
Schematic drawings of a healthy retina and a inherited retinal degeneration (IRD) retina. (**A**) Illustration of the various layers of an intact, healthy retina, from the retinal pigment epithelium (RPE) to the ganglion cell layer. Rod photoreceptors in the outer retina are shown in black, while cones are indicated by red, green, and blue. (**B**) Degenerated IRD retina. The outer nuclear layer is almost completely lost and the outer plexiform layer has essentially disappeared. Curiously, when the retina has lost all functionality, a small number of cone photoreceptors may still be present, possibly for many years beyond the loss of rod photoreceptors. BC = bipolar cell; GC = ganglion cell; MC = Müller glial cell. Note that the retinal structure has been simplified for clarity and that not all retinal cell types are shown.

**Figure 2 genes-10-00453-f002:**
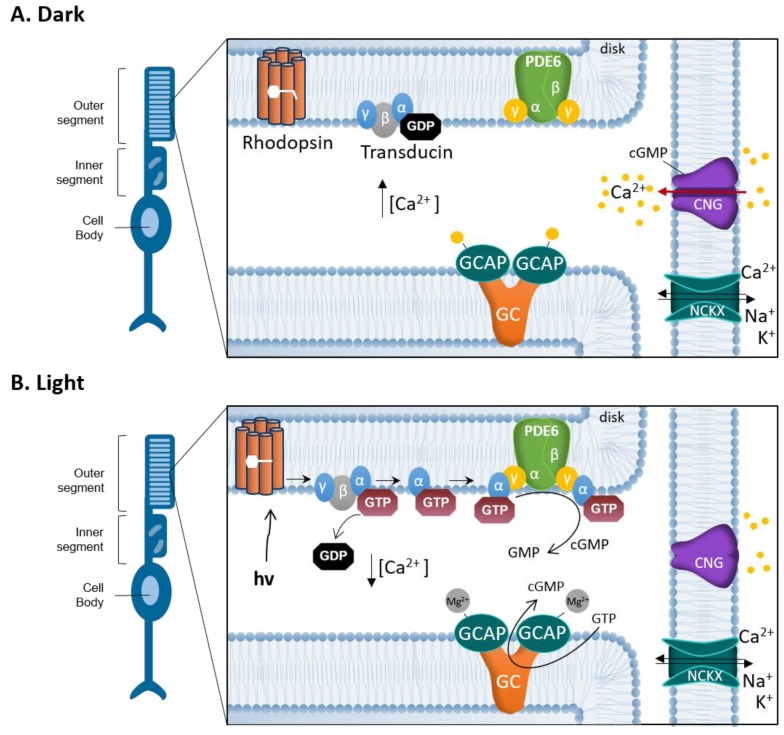
Phototransduction and the photoreceptor cGMP-Ca^2+^ feedback loop. Schematic representation of the interplay between cGMP and Ca^2+^ in the photoreceptor outer segment (OS). (**A**) In darkness, cGMP binds to the cyclic nucleotide-gated channel (CNGC). The opening of CNGC allows for an influx of Na^+^ and Ca^2+^ into the photoreceptor OS. At the same time K^+^ and Ca^2+^ ions are constantly extruded via Na^+^/Ca^2+^/K^+^ exchanger (NCKX) creating a continuous influx and outflow of ions called the dark current. Ca^2+^ binds GC activating proteins (GCAPs), which inhibit the synthesis of cGMP by limiting guanylyl cyclase (GC) activity. (**B**) In light, photon (hν) absorption induces conformational changes in the rhodopsin protein. Rhodopsin stimulates the GTP-binding protein transducin to detach from heteromeric G-protein complex, by replacing bound GDP with GTP. The activated transducin α subunit binds to the PDE6 complex, abolishing the inhibitory effect exerted by its γ subunits. Activated phosphodiesterase-6 (PDE6) hydrolyses cGMP to GMP, which in turn limits the CNGC opening and leads to a reduction of Ca^2+^ influx. The closure of the CNGC and a hyperpolarization of the OS, due to continued activity of NCKX, promote the generation of an electro-chemical signal that is transmitted to second order neurons. When OS Ca^2+^ concentration is reduced, Mg^2+^ replaces the Ca^2+^ bound to GCAP, reactivating GCAP and stimulating GC to synthesize cGMP, opening the CNGC again.

**Figure 3 genes-10-00453-f003:**
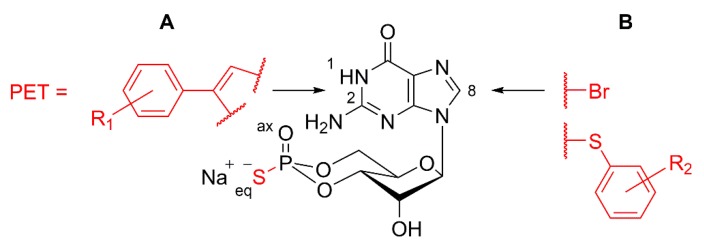
Schematic structure of Rp-cGMPS analogues. Arrows illustrate typical structural positions in Rp-cGMPS used for the introduction of substituents to generate Rp-cGMPS analogues with improved inhibitory potency for cGMP-dependent protein kinase I or II. (**A**) Position 1, N^2^: Addition of β-phenyl-1,N^2^-etheno-modifications (PET) with varying additional substituents (R_1_) to Rp-cGMPS [65,66]. (**B**) Position 8: Addition of halogens (e.g., bromine (Br)) or sulfur-connected aromatic ring systems with varying additional substituents (R_2_) [65,67].

**Figure 4 genes-10-00453-f004:**
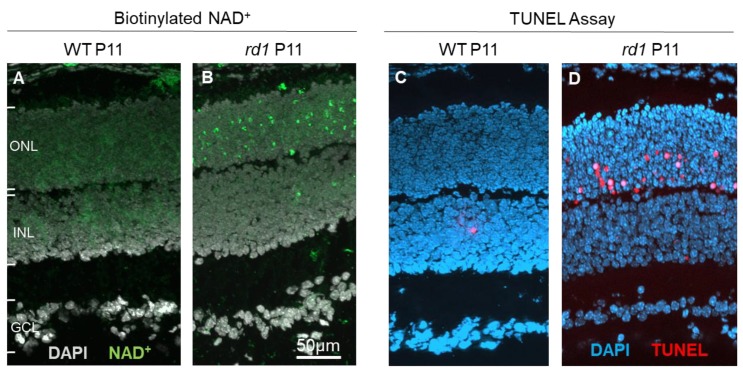
Incorporation of NAD^+^ as a biomarker for retinal cell death. At post-natal day 11 (P11), when compared to wild-type (WT) retina (**A**), photoreceptors in the *rd1* mouse model (**B**) display marked incorporation of biotinylated NAD^+^. This is highly correlated with the TUNEL assay for cell death, which at the same age detects only few cells in the WT situation (**C**), while large numbers are detected in the *rd1* outer nuclear layer (ONL; **D**). INL = inner nuclear layer, GCL = ganglion cell layer.

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
