# Peer review of "The cGMP Pathway and Inherited Photoreceptor Degeneration: Targets, Compounds, and Biomarkers"

_genes, 2019, doi:10.3390/genes10060453_

Reviewer 1 Report

The article is a review of the importance of cGMP-signaling for the pathogenesis of inherited retinal degeneration. In my opinion, adding more relevant information to this article will improve it and help readers to better understand the subject.

It would be useful if the authors could add some background sentences and/or hypotheses about all the underlying mechanisms, not just the genetic ones (e.g. oxidative stress, trophic factors, etc) of rod and cone degeneration in the introduction, as well as to extent the explanation of the consequences for the inner retina (retinal remodeling) that probably ends with the loss of retinal ganglion cells (afferent neurons).

It would be interesting to introduce the main types of RD and the diverse genes mutation in these patients, onset of disease, progression and prevalence.

The authors stated that when “PKG II was knocked-out in mice, … showed a defect in the entrainment of circadian rhythm”. This could mean that such an inhibition may affect intrinsically photosensitive retinal ganglion cells. The authors should clarify this fact and if this inhibition may affect the general RGC populations considering the possible consequences.

The authors may want to expand the “DDS for the retina section” to further clarify which administration routes are most commonly used and their most common risks.

The side effects of repeated intravitreal injections should also be considered for local administration.

The manuscript contains various spelling and grammar errors and should be proofread and corrected by a person proficient in the English language. Authors should also pay attention to the use of abbreviations (for example see the use of outer segments (OS) in lines 72 and 74).

Author Response

Reviewer 1:

The article is a review of the importance of cGMP-signalling for the pathogenesis of inherited retinal degeneration. In my opinion, adding more relevant information to this article will improve it and help readers to better understand the subject. It would be useful if the authors could add some background sentences and/or hypotheses about all the underlying mechanisms, not just the genetic ones (e.g. oxidative stress, trophic factors, etc.) of rod and cone degeneration in the introduction, as well as to extent the explanation of the consequences for the inner retina (retinal remodelling) that probably ends with the loss of retinal ganglion cells (afferent neurons). It would be interesting to introduce the main types of RD and the diverse genes mutation in these patients, onset of disease, progression and prevalence.

Reply: In principle - if this was a review for general journal - we would agree with the reviewer and the idea of introducing the diseases in more detail. However, since this article is intended for a thematic issue on hereditary diseases of the retina, we feel that more details on the diseases would likely be redundant. Of course, if the editors also felt that such additional information would still be needed in the context of the thematic issue, we would be prepared to add this.

The authors stated that when “PKG II was knocked-out in mice, … showed a defect in the entrainment of circadian rhythm”. This could mean that such an inhibition may affect intrinsically photosensitive retinal ganglion cells. The authors should clarify this fact and if this inhibition may affect the general RGC populations considering the possible consequences.

Reply: We thank the reviewer for this interesting comment. In Oster et al. 2003, the authors show that cGKII−/− mice are defective in resetting the circadian clock. However, it remains unclear whether this leads to changes in the photosensitive retinal ganglion cells. We rephrased the sentence in the revised manuscript in order to better clarify this concept (lines 167-170).

The authors may want to expand the “DDS for the retina section” to further clarify which administration routes are most commonly used and their most common risks. The side effects of repeated intravitreal injections should also be considered for local administration.

Reply: We thank the reviewer for this suggestion and have expanded the chapter 5a accordingly (lines 339-349). The side effects of repeated intravitreal injections, notably intraocular inflammation but possibly also systemic effects, are indeed a concern to which we also refer in chapter 5b (lines 365-370).

The manuscript contains various spelling and grammar errors and should be proofread and corrected by a person proficient in the English language. Authors should also pay attention to the use of abbreviations (for example see the use of outer segments (OS) in lines 72 and 74).

Reply: We thank the reviewer for this comment and accordingly have thoroughly revised the manuscript for orthographic or grammatical mistakes and also revised the use of abbreviations.

Reviewer 2 Report

Manuscript : The cGMP pathway and inherited photoreceptor degeneration: Targets,

compounds, and biomarkers

This manuscript summarizes the current knowledge about cGMP pathway role in hereditary retinal degeneration. The authors chose a topic that could be of high interest to readers concerned with retinal degeneration which encompasses a number of conditions that can lead to vision impairment or even loss. Covering such topic would draw the attention of those interested in the field to the challenges associated with developing an effective treatment and the need to look for more innovative approaches.  The general organization and choice of themes is good. However, there is room for improvement that could be achieved by incorporating and addressing the points below.

Lines 32-33 “making RD a major unmet  medical problem” the use of the word “unmet” to described a problem  in this context is unusual a better alternative is (unresolved)

Figure 1 (B) the cone cell in the outer nuclear layer is in orange instead of red.

Lines 60-62 “This enormous heterogeneity calls for the development of treatment approaches that target common principles, which can be used to address many different disease causing mutations at once”

 The sentence needs to be revised; (common principles) an uncommon use of the word principles to describe biological pathways and mechanisms. For the sake of clarity the authors could use standard words (mechanism or pathways).

“which can be used to address many different disease causing mutations at once”

This sentence may imply the use of gene editing for therapeutic application, which is not part of the scope of this review, therefore this sentence could be re-phrased to read something like (This extensive heterogeneity calls for the development of therapeutic approaches targeting common pathways, that can effectively treat this condition regardless of the underlying genetic causes)

Lines 71-83 the authors can include an illustrative diagram to aid the understanding of the well described phototransduction cascade

Lines 84-85 “Seminal research by Deborah Farber and Frank Lolley established the toxicity of high levels of cGMP for photoreceptors already in the 1970s “

Revisit the construction of the sentence, “already” is placed at the end of the sentence when usually it should be at a mid-position. A suggested modification (The toxicity of high levels of cGMP for photoreceptors was already established by Deborah Farber and Frank Lolley  in the 1970s).

Lines 89-90 “However, cGMP elevation was also found in mutations occurring in genes that seem to have no obvious connection to cGMP metabolism”.

Proteins or signaling pathways are not expressed in genes and certainly not in their mutations, this sentence is inaccurate. I am assuming the authors meant to say (However, cGMP elevation was also found in tissues/cells with mutations in genes that seem to have no obvious connection to cGMP metabolism)

Figure 2 could be made more effective for helping the readers understand the complicated pathway by considering the following modifications:

1)      Indicate outer and inner segments

2)      PDE6 hydrolyse in the figure is PDE instead of PDE6

3)      Enlarge the font/shape size for GDP and GTP molecules to make them more clearly visible

In the light phase Figure 2B:

1)      Light-induced conformational changes to Rhodopsin are not reflected in the illustration

2)      Rhodopsin activating effect on Transducin can be indicated using a sold arrow

(Rhodopsin                           Transducin)

3)      The sequential steps for transducin activation could be broken down into:

a)      Replacement of GDP with GTP

b)      Translocation of α-transducin subunit to PDE6

Line 142 “kinase that phosphorylates a number of biologically targets”

The typo should be corrected to (biological)

Lines 157-158 “Interestingly, this effect could be rescued by a knock-in of PKG I in smooth muscles [43]”

In knock-in experiments a new (foreign) genetic material is introduced to the organism, however, when a gene- that is normally part of the organism’s genome- is re-introduced into the a transgenic model lacking it (knock out) the more accurate term to describe this manipulation is re-constitution. In the study cited in this sentence [ref 43] Weber and his colleagues reconstituted their mice with genes encoding the murine cGKIα or -Iβ isozymes not the human (PKG I). The sentence here conveys inaccurate information.

Lines 159-160 “In sensory cells of the inner ear, a  knock-out of PKG I did not affect hearing function in detectable ways, yet, PKG I knock-out animals were protected against acoustic trauma and resistant against noise induced hearing loss”

Same as above, the sentence here conveys inaccurate information. The authors of the cited study knocked out the murine version not the human version of the gene therefore PKG I should be replaced with PrkgI.

Lines 161-162 “When  PKG II was knocked-out in mice, the animals appeared generally healthy but showed a defect in the  entrainment of circadian rhythm, even though retinal morphology and function appeared normal”

Same as above, the sentence here conveys inaccurate information. The authors of the cited study knocked out the murine version not the human version of the gene therefore PKG II should be replaced with cGKII.

Lines 163-164 “Overall, the results obtained on PKG knock-out animals [37] suggest that targeted inhibition of PKG  would be unlikely to produce major adverse effect”

This sentence needs to be revised. First, only genes but not proteins are manipulated in transgenic animals (knockout or knock in), therefore “PKG knock-out animals” could be replaced with (animals deficient for PKG enzymes) this includes animals lacking all PKG isoforms due to deleting the corresponding genes.

“would be unlikely to produce major adverse effect” revise sentence syntax (would unlikely produce major adverse effect)

Lines 171-172 “This evidence makes PKG a potential target for neuroprotective strategies for the prevention of neuronal cell death.”

The sentence could be re-written to omit redundancy (neuroprotective…. prevention of neuronal cell death), suggested re-phrasing (This evidence makes PKG a potential target for neuroprotective strategies.)

Line 182 “NO/PKG/extracellular-signal-regulated kinases (ERK) signalling mediated the PKG-inhibitor KT5823”

 The word (by) is missing (NO/PKG/extracellular-signal-regulated kinases (ERK) signalling mediated by the PKG-inhibitor KT5823….)

Lines 186-187 “However, it is also a weak inhibitor of PKC and PKA, which can constitute potential competitive binding sites [56,57].”

Do PKC and PKA both have competitive binding sites for KT5823?

If yes, then the sentence could be modified to clearly state this (However, it is also a weak inhibitor of PKC and PKA, as both of which may constitute potential competitive binding sites)

Lines 189-190 “Furthermore, in whole cell homogenates the PKG 189 specificity over PKA of DT-2 appear to be lost [58].”

This sentence has some typo/grammar errors, it should be modified to (Furthermore, in whole cell homogenates the PKG  specificity over PKA for DT-2 appears to be lost)

Lines 196-197 “Over the last ~40  years, earlier cGMP analogues have become standard tools for investigations of biochemical and….”

Two words describing period of time are used here consecutively, which is confusing for the reader. I suggest omitting the word “earlier”.

Line 340 “synchronize the subsequent and necessary compound and delivery development”

I am not sure I understand what the authors meant to say here?

Lines 342-343 “Currently, the standard route of administration for retinal drugs that will be applied multiple times, such as anti-VEGF medications, is intravitreal injection [89,91],..”

The construction of the sentence could be improved to read something like (Currently, the standard route of administration for retinal drugs, such as anti-VEGF medications, is multiple dosing through intravitreal injection….)

Lines 343-353 “Unfortunately, topical drug application is usually not possible because the various  barriers surrounding the eye and retina, and the alternating lipophilic and hydrophilic nature of these”

The sentence has two typos:

 The word (of) is missing (application is usually not possible because  of  the….)

The tense for the verb “prevents” is not correct (preventing most drugs from reaching the retina via this route)

Conclusion and outlook

It would be very relevant and informative if the authors could add a paragraph addressing the following:

1)    Did the Rp-cGMPS analogues move to clinical trial testing if not why?

2)    Are there other cGMP/S analogues?  

3)    What are the limitations of using cGMP targeted therapies? for example, would they be effective in  RD forms that do not involve cGMP signaling dysfunction?

4)    Would combination therapy hold promise of improved treatment?

Author Response

Reviewer 2:

This manuscript summarizes the current knowledge about cGMP pathway role in hereditary retinal degeneration. The authors chose a topic that could be of high interest to readers concerned with retinal degeneration which encompasses a number of conditions that can lead to vision impairment or even loss. Covering such topic would draw the attention of those interested in the field to the challenges associated with developing an effective treatment and the need to look for more innovative approaches.  The general organization and choice of themes is good. However, there is room for improvement that could be achieved by incorporating and addressing the points below.

Lines 32-33 “making RD a major unmet  medical problem” the use of the word “unmet” to described a problem  in this context is unusual a better alternative is (unresolved)

Reply: We thank the reviewer for this suggestion and have changed the wording accordingly (line 34).

Figure 1 (B) the cone cell in the outer nuclear layer is in orange instead of red.

Reply: We have changed the colour of the red cone in Figure 1 (B) to match the colour used in Figure 1 (A).

Lines 60-62 “This enormous heterogeneity calls for the development of treatment approaches that target common principles, which can be used to address many different disease causing mutations at once”. The sentence needs to be revised; (common principles) an uncommon use of the word principles to describe biological pathways and mechanisms. For the sake of clarity, the authors could use standard words (mechanism or pathways). “which can be used to address many different disease causing mutations at once”. This sentence may imply the use of gene editing for therapeutic application, which is not part of the scope of this review, therefore this sentence could be re-phrased to read something like (This extensive heterogeneity calls for the development of therapeutic approaches targeting common pathways, that can effectively treat this condition regardless of the underlying genetic causes)

Reply: We have changed the wording as suggested (lines 61-62).

Lines 71-83 the authors can include an illustrative diagram to aid the understanding of the well described phototransduction cascade

Reply: We appreciate this suggestion, although Figure 2 in principle shows the main components of the phototransduction cascade already. Therefore, we have now revised Figure 2 and its legend to better illustrate phototransduction as well.

Lines 84-85Seminal research by Deborah Farber and Frank Lolley established the toxicity of high levels of cGMP for photoreceptors already in the 1970s “. Revisit the construction of the sentence, “already” is placed at the end of the sentence when usually it should be at a mid-position. A suggested modification (The toxicity of high levels of cGMP for photoreceptors was already established by Deborah Farber and Frank Lolley in the 1970s).

Reply: We have changed the wording accordingly (lines 83-84).

Lines 89-90 “However, cGMP elevation was also found in mutations occurring in genes that seem to have no obvious connection to cGMP metabolism”. Proteins or signalling pathways are not expressed in genes and certainly not in their mutations, this sentence is inaccurate. I am assuming the authors meant to say (However, cGMP elevation was also found in tissues/cells with mutations in genes that seem to have no obvious connection to cGMP metabolism).

Reply: We thank the reviewer for this suggestion and have changed the wording accordingly.

Figure 2 could be made more effective for helping the readers understand the complicated pathway by considering the following modifications:
1)      Indicate outer and inner segments
2)      PDE6 hydrolyse in the figure is PDE instead of PDE6
3)      Enlarge the font/shape size for GDP and GTP molecules to make them more clearly visible

In the light phase Figure 2B:
1)      Light-induced conformational changes to Rhodopsin are not reflected in the illustration
2)      Rhodopsin activating effect on Transducin can be indicated using a sold arrow
→(Rhodopsin                           Transducin)
3)      The sequential steps for transducin activation could be broken down into:
a)      Replacement of GDP with GTP
b)      Translocation of α-transducin subunit to PDE6

Reply: We thank the reviewer for this suggestion. As mentioned above, we have now revised Figure 2 and its legend to better illustrate this pathway.

Line 142 “kinase that phosphorylates a number of biologically targets” The typo should be corrected to (biological)

Reply: We apologize for this mistake and have corrected this in the revised manuscript (line 148).

Lines 157-158 “Interestingly, this effect could be rescued by a knock-in of PKG I in smooth muscles [43]” In knock-in experiments a new (foreign) genetic material is introduced to the organism, however, when a gene- that is normally part of the organism’s genome- is re-introduced into a transgenic model lacking it (knock out) the more accurate term to describe this manipulation is re-constitution. In the study cited in this sentence [ref 43] Weber and his colleagues reconstituted their mice with genes encoding the murine cGKIα or -Iβ isozymes not the human (PKG I). The sentence here conveys inaccurate information.

Reply: We thank the reviewer for this suggestion. We rephrased the sentence accordingly (lines 163-165).

Lines 159-160 “In sensory cells of the inner ear, a knock-out of PKG I did not affect hearing function in detectable ways, yet, PKG I knock-out animals were protected against acoustic trauma and resistant against noise induced hearing loss” Same as above, the sentence here conveys inaccurate information. The authors of the cited study knocked out the murine version not the human version of the gene therefore PKG I should be replaced with Prkg1.

Reply: We changed the gene symbol accordingly throughout subchapter 3b (lines 147-179).

Lines 161-162 “When PKG II was knocked-out in mice, the animals appeared generally healthy but showed a defect in the entrainment of circadian rhythm, even though retinal morphology and function appeared normal” Same as above, the sentence here conveys inaccurate information. The authors of the cited study knocked out the murine version not the human version of the gene therefore PKG II should be replaced with cGKII.

Reply: We now refer to the murine genes as Prkg1 or Prkg2 (lines 147-179).

Lines 163-164 “Overall, the results obtained on PKG knock-out animals [37] suggest that targeted inhibition of PKG would be unlikely to produce major adverse effect” This sentence needs to be revised. First, only genes but not proteins are manipulated in transgenic animals (knockout or knock in), therefore “PKG knock-out animals” could be replaced with (animals deficient for PKG enzymes) this includes animals lacking all PKG isoforms due to deleting the corresponding genes.
“would be unlikely to produce major adverse effect” revise sentence syntax (would unlikely produce major adverse effect).

Reply: We thank the reviewer for this suggestion and have changed the wording accordingly (lines 171-172).

Lines 171-172 “This evidence makes PKG a potential target for neuroprotective strategies for the prevention of neuronal cell death.” The sentence could be re-written to omit redundancy (neuroprotective…. prevention of neuronal cell death), suggested re-phrasing (This evidence makes PKG a potential target for neuroprotective strategies.)

Reply: We have taken over this suggestion from the reviewer (line 179).

Line 182 “NO/PKG/extracellular-signal-regulated kinases (ERK) signalling mediated the PKG-inhibitor KT5823” The word (by) is missing (NO/PKG/extracellular-signal-regulated kinases (ERK) signalling mediated by the PKG-inhibitor KT5823….).

Reply: We apologize for this mistake and have corrected this in the revised manuscript (line 189).

Lines 186-187 “However, it is also a weak inhibitor of PKC and PKA, which can constitute potential competitive binding sites [56,57].” Do PKC and PKA both have competitive binding sites for KT5823?
If yes, then the sentence could be modified to clearly state this (However, it is also a weak inhibitor of PKC and PKA, as both of which may constitute potential competitive binding sites).

Reply: We have changed the wording accordingly (lines 193-195).

Lines 189-190 “Furthermore, in whole cell homogenates the PKG 189 specificity over PKA of DT-2 appear to be lost [58].” This sentence has some typo/grammar errors, it should be modified to (Furthermore, in whole cell homogenates the PKG specificity over PKA for DT-2 appears to be lost).

Reply: We have corrected this in the revised manuscript (lines 196-197).

Lines 196-197 “Over the last ~40 years, earlier cGMP analogues have become standard tools for investigations of biochemical and….” Two words describing period of time are used here consecutively, which is confusing for the reader. I suggest omitting the word “earlier”.

Reply: We have changed this as suggested (line 204).

Line 340 “synchronize the subsequent and necessary compound and delivery development”
I am not sure I understand what the authors meant to say here?

Reply: We have simplified this sentence to improve its understandability (lines 359-360).

Lines 342-343 “Currently, the standard route of administration for retinal drugs that will be applied multiple times, such as anti-VEGF medications, is intravitreal injection [89,91],…” The construction of the sentence could be improved to read something like (Currently, the standard route of administration for retinal drugs, such as anti-VEGF medications, is multiple dosing through intravitreal injection….)

Reply: We thank the reviewer for this suggestion and have changed the wording accordingly (lines 362-363).

Lines 343-353 “Unfortunately, topical drug application is usually not possible because the various barriers surrounding the eye and retina, and the alternating lipophilic and hydrophilic nature of these” The sentence has two typos: -The word (of) is missing (application is usually not possible because  of  the….); -The tense for the verb “prevents” is not correct (preventing most drugs from reaching the retina via this route).

Reply: We have corrected this in the revised manuscript (lines 372-374).

Conclusion and outlook

It would be very relevant and informative if the authors could add a paragraph addressing the following:

1)    Did the Rp-cGMPS analogues move to clinical trial testing if not why?

Reply: This is indeed a very interesting question from the reviewer. One could argue that cGMP generating drugs have been marketed for decades (e.g. nitroglycerol, Viagra). For the clinical development of cyclic nucleotide analogues, an important problem in the past has been the rapid clearance via the kidney; the compounds simply do not stay long enough in the body to be biologically active. This makes it essential to use an appropriate DDS, something that we now mention this in chapter 5b (lines 377-380). With this DDS technology now at hand, we are ourselves actively pursuing the clinical development of cGMP analogues, however, it may be several years still until one can say how successful this may be.

2)    Are there other cGMP/S analogues?

Reply: As indicated in the schematic overview in Figure 3 the possibilities to generate other cGMP analogues are almost limitless. As a matter of fact, the DRUGSFORD project alone generated about 85 inhibitory cGMPS analogues (along with around 100 activatory analogues). This information has now been added to the text (line 226).

3)    What are the limitations of using cGMP targeted therapies? for example, would they be effective in RD forms that do not involve cGMP signalling dysfunction?

Reply: We would consider this unlikely and – at least for clinical development – focus on patients with gene mutations that are clearly connected to elevated photoreceptor cGMP levels. However, there are also situations (albeit seemingly very rare) in which photoreceptor cGMP levels may be too low. We now address this point in the conclusion and outlook (lines 391-394)

4)    Would combination therapy hold promise of improved treatment?

Reply: Combination therapy to try and harvest synergistic beneficial effects of different compounds is an exciting perspective. Indeed, in fast-progression, lethal diseases, especially in cancer, combination therapy has had remarkable success in the past 30 years. However, in slow progression neurodegenerative diseases both pre-clinical and clinical development of combination therapies is extremely complicated to pursue, notably because the number of tests to perform increases exponentially with the number of compounds to be combined, and the regulatory safety requirements are difficult to fulfil. Therefore, and since we are not aware of any such developments under way, we would prefer not to discuss combination therapy in this review.

Round  2

Reviewer 1 Report

The manuscript has been significantly improved and I feel that it is now suitable for publication.

Author Response

The manuscript has been significantly improved and I feel that it is now suitable for publication.

Reply: We thank the reviewer for this comment.